# Protective Potential of *Satureja montana*-Derived Polyphenols in Stress-Related Central Nervous System Disorders, Including Dementia

**DOI:** 10.3390/cimb47070556

**Published:** 2025-07-17

**Authors:** Stela Dragomanova, Lyubka Tancheva, Silviya Abarova, Valya B. Grigorova, Valentina Gavazova, Dana Stanciu, Svetlin Tzonev, Vladimir Prandjev, Reni Kalfin

**Affiliations:** 1Department of Pharmacology, Toxicology and Pharmacotherapy, Faculty of Pharmacy, Medical University, 9002 Varna, Bulgaria; 2Institute of Neurobiology, Bulgarian Academy of Sciences, 23 Academician Georgi Bonchev St, 1113 Sofia, Bulgaria; lyubkatancheva@gmail.com (L.T.); vbgrigorova@abv.bg (V.B.G.); 3Department of Medical Physics and Biophysics, Faculty of Medicine, Medical University of Sofia, 15 Academician Ivan Evstratiev Geshov Blvd., 1431 Sofia, Bulgaria; silviya.m.abarova@gmail.com; 4Clinic of Endocrinology, Military Medical Academy, 3 Sveti Georgi Sofiyski St, 1606 Sofia, Bulgaria; dr.vgavazova@gmail.com; 5Department of Pharmaceutical Botany, Faculty of Pharmacy, Iuliu Hațieganu University of Medicine and Pharmacy, 400337 Cluj-Napoca, Romania; dana.stanciu@elearn.umfcluj.ro; 6Faculty of Medicine, University Prof. Assen Zlatarov, 1 Prof. Yakimov St., 8010 Burgas, Bulgaria; svetmed@gmail.com; 7Department of Neurosurgery, Military Medical Academy, 3 Sveti Georgi Sofiyski St, 1606 Sofia, Bulgaria; prandjev@hotmail.com; 8Department of Healthcare, Faculty of Public Health, Healthcare and Sport, South-West University, 66 Ivan Mihailov St, 2700 Blagoevgrad, Bulgaria

**Keywords:** *Satureja montana*, polyphenols, cognitive impairment, dementia, Alzheimer’s disease

## Abstract

*Satureja* *montana* (SM) is acknowledged as a highly pharmacologically important species within the vast Lamiaceae family, indigenous to the Balkan area. Traditionally, this plant has been employed as a culinary spice, especially in Bulgarian gastronomy. Additionally, it is widely recognized that mental health is affected by the nature and quality of dietary consumption. Results: Ethnopharmacological research underscores the potential of SM in influencing various chronic ailments, including depression and anxiety. This plant is distinguished by a rich variety of secondary metabolites that display a broad spectrum of biological activities, such as antioxidant, antidiabetic, anti-inflammatory, analgesic, antibacterial, antiviral, and antifungal effects. Particularly, two of its active phenolic compounds, rosmarinic acid and carvacrol, reveal notable anxiolytic and antidepressive properties. This review aims to explore the capacity of SM to improve mental health through its plentiful phenolic components. Recent studies indicate their efficacy in addressing Alzheimer’s-type dementia. A notable correlation exists among depression, anxiety, and cognitive decline, which includes dementia. Considering that Alzheimer’s disease (AD) is a multifaceted condition, it requires multi-targeted therapeutic strategies for both prevention and management. Conclusions: *Satureja montana* is recognized as potential candidate for both the prevention and management of various mental health disorders, including dementia.

## 1. Introduction

Historically, medicinal and herbal plants have been utilized to treat a wide range of ailments and to alleviate symptoms of diseases. Despite significant progress in pharmaceutical production, it is estimated that 80–90% of the global population still relies on traditional medicine for treatment [1]. According to the World Health Organization, 80% of people worldwide use herbal remedies, with the market valued at approximately USD 83 billion in 2019 and projected to grow to USD 550 billion by 2030 [2]. Nevertheless, even with the increasing interest from the pharmaceutical industry in naturally sourced compounds [3], only a limited number of the therapeutic properties and biological activities of medicinal plants have been extensively studied [4,5,6]. The exploration of a medicinal plant’s potential as a therapeutic agent is inherently intricate, necessitating thorough investigations that cover its application, cultivation, extraction, identification of active compounds, efficacy, safety, and clinical evaluation [7,8,9]. While various parts of medicinal plants, including roots, seeds, leaves, and fruits, have been historically acknowledged as sources of bioactive compounds and natural products (such as salicylates, digitalis, and quinine), the transition from natural sources to drug development continues to be a demanding and labor-intensive endeavor.

Plants rich in polyphenols hold significant relevance in ethnopharmacology due to their diverse biological activities and health advantages. Various cultures integrate these plants into their traditional healing methods [10,11,12]. Polyphenols are especially noted for their strong antioxidant properties, which help neutralize free radicals and protect cells from oxidative harm. These attributes are crucial in preventing chronic diseases such as cardiovascular issues, cancer, and neurodegenerative disorders [13,14]. Moreover, polyphenols exhibit considerable anti-inflammatory effects, beneficial for managing conditions like arthritis and other inflammatory diseases. For instance, extracts from *Rhamnus prinoides* have shown substantial anti-inflammatory properties [15]. Additionally, polyphenols possess antimicrobial capabilities, making them effective against a range of pathogens, including bacteria, fungi, and viruses. This quality is particularly advantageous in traditional medicine for treating infections [16,17,18]. The Lamiaceae family comprises more than 230 genera and over 7000 species [19], which include thyme, mint, lemon balm, oregano, basil, rosemary, sage, and Balkan savory [20,21,22,23,24,25]. *Satureja montana* (Figure 1), commonly referred to as Balkan savory, is recognized as one of the most pharmacologically significant members of the extensive Lamiaceae family [26]. Balkan savory originates from the Mediterranean area. Although the present distribution of *Satureja montana* has expanded beyond its native Mediterranean habitat, it is not regarded as invasive in any ecosystems.

*Satureja montana* is widely distributed across the Balkan Peninsula and is considered an endemic species in Bulgaria [23,24,25]. Research indicates that extracts from SM are rich in total polyphenols and flavonoids, which play a significant role in its diverse biological functions [27]. Notable phenolic compounds such as rosmarinic acid, caffeic acid, and luteolin have been identified in substantial amounts within various extracts [28,29,30]. These phenolic constituents are responsible for the extensive range of biological activities exhibited by SM, including its antimicrobial and antioxidant effects.

For centuries, the plant has been recognized as a flavorful spice, particularly in traditional Bulgarian cuisine. This plant is not only versatile but also holds considerable significance in ethnopharmacology. The ethnobotanicity index quantifies the level of traditional knowledge and utilization of plants within a community. Balkan savory exhibits a high ethnobotanicity index in areas where it is customarily employed for medicinal purposes [31,32]. Additionally, the fidelity level index reflects the proportion of informants who report using a specific plant for the same primary purpose. Balkan savory demonstrates elevated fidelity level values in the treatment of digestive and respiratory ailments. In traditional medicine, it is utilized for various health conditions. *Satureja montana* is characterized by a diverse array of secondary metabolites that exhibit a wide range of biological activities, including antioxidant [28,30,33,34], antidiabetic [22,35], anti-inflammatory [33,36], neuroprotective [37,38,39,40], analgesic, antibacterial, antiviral, and antifungal properties [31,41,42,43,44].

The objective of the current review is to explore the potential of SM, both as a medicinal herb and a traditional spice, in promoting mental health through its rich array of phenolic compounds. It is well-established that mental health is affected by the quality and type of food consumed. Recent studies are increasingly clarifying the connection between dietary practices and mental well-being, with a multitude of research indicating a significant correlation between the two [45]. The functionality, structure, and composition of the brain are contingent upon the availability of vital amino acids, lipids, vitamins, and minerals, making cognitive performance and mental health susceptible to nutritional variations [46,47,48,49]. Evidence suggests that following a nutritious diet, characterized by a high consumption of spices, vegetables, fruits, fish, nuts, and whole grains—similar to the Mediterranean diet—can have positive effects, protect mental health, and reduce the risk of depression [50,51,52,53]. Furthermore, research into the effects of poor dietary choices on mental health and cognitive functions highlights the importance of sustaining a healthy diet [54,55,56,57,58]. Additionally, factors such as neurotransmitters, neuropeptides, endogenous gut hormones, and gut microbiota can also be influenced by changes in dietary composition [59,60,61,62,63].

## 2. Materials and Methods

To accomplish the established goal, we gathered and thoroughly examined the most recent scientific literature regarding the neuroprotective properties of *Satureja montana* extracts and their primary polyphenolic constituents. A comprehensive literature review was conducted utilizing the Web of Science, Scopus, PubMed, and ResearchGate databases, from which we selected a number of articles, predominantly published within the last decade. The research employed a variety of keywords and their combinations, including SM, polyphenols, rosmarinic acid, ellagic acid, chlorogenic acid, antioxidant, neuroprotection, neurotransmitter, oxidative stress, inflammation, apoptosis, anti-cholinesterase, and signaling pathways. The selection process entailed the removal of duplicate entries, screening of titles and abstracts, and a detailed full-text assessment of potentially pertinent articles. Both original research and review articles published in English were incorporated into the manuscript.

The inclusion criteria are articles published in peer-reviewed journals; studies published in English; original research and review articles that concentrate on the neuroprotective, antioxidant, anti-inflammatory, or cholinesterase-inhibiting properties of *Satureja montana* extracts or their individual polyphenols; and studies that include in vitro, in vivo, or clinical data.

Conversely, the criteria for exclusion are non-English publications; articles lacking sufficient or relevant data concerning neuroprotective activity; publications that consist solely of abstracts, conference proceedings, or editorials; and studies that exclusively examine other species without a specific mention of *Satureja montana* or its pertinent constituents.

The screening strategy included the elimination of duplicate entries using reference management software. The remaining records underwent a two-step screening procedure: (1) an initial review of titles and abstracts for relevance, followed by (2) a full-text assessment of the articles that remained. Two authors conducted the screening independently, and any discrepancies were addressed through discussion.

To evaluate the quality of the studies included, we took into account factors such as the clarity of the study design, the suitability of experimental models, the robustness of statistical analysis, and the relevance to neuroprotective mechanisms. For both in vivo and in vitro studies, we assessed parameters like sample size, the inclusion of control groups, and the reproducibility of results. Review articles were evaluated based on their thoroughness and the citation of primary literature.

## 3. Results

### 3.1. Bioactive Compounds of Satureja montana

The main compounds derived from *Satureja montana* can be categorized into several classes based on their chemical nature. From the group of monoterpenes, carvacrol, p-cymene, γ-terpinene, and thymol are found [64,65]. Oxygenated monoterpenes are represented by carvacrol methyl ether, borneol, and linalool [66]. Of the sesquiterpenes, β-Caryophyllene is present in concentrations ranging from 2.74% to 4.71%, and Spathulenol is found in smaller amounts [67]. The main phenolic acid components isolated from savory are rosmarinic and ellagic acids [28]. In addition, an investigation examining the composition of two varietal forms of *Satureja montana* observed that the phenolic compounds’ carvacrol:thymol ratio was significantly elevated in the subsp. *montana* (exceeding 300) in comparison to the subsp. *variegate* [65]. 

The primary constituents of the essential oil include carvacrol, p-cymene, γ-terpinene, and thymol. Carvacrol is the most abundant phenolic monoterpene, with concentrations ranging from 44.5% to 45.7% [68,69]. P-cymene accounts for approximately 12.6% to 16.9%, while γ-terpinene is present in amounts between 8.1% and 8.7% [68,69]. Thymol is also present in considerable quantities, reaching 81.79% in certain studies [26]. In terms of phenolic compounds, rosmarinic acid is recognized as a significant phenolic component in the dry extract of *Satureja montana* [29]. Additional phenolic acids identified in SM include caffeic acid, chlorogenic acid, and ellagic acid [26,28,70]. The triterpenes compounds are represented by ursolic and oleanolic acids [70].

Table 1 presents a summary of the constituents isolated from Balkan savory.

*Satureja montana* possesses a high concentration of phenolic compounds, which makes the plant an important resource for both medicinal applications and preservation purposes.

Two SM active phenolic compounds—rosmarinic acid and carvacrol—possess strong anxiolytic and antidepressive effects [71,72]. Each of these two phenolic compounds also exhibits antiviral, antibacterial, antioxidant, antimutagenic, anti-inflammatory, and anxiolytic effects [73,74,75,76,77,78].

A summary of the available data is presented in Table 2.

A comprehensive study on *Satureja montana* was recently conducted in Bulgaria [79]. The dry methanol–water extract of SM was standardized based on its rosmarinic acid and carvacrol content. The findings indicated that the total phenolic content of the dry extract of SM was greater than that reported in the previous literature.

### 3.2. Relation of Stress to the Pathogenesis of Certain Mental Disorders

Stress, as a variable significantly shaped by lifestyle choices, in addition to being a physiological defense response of the body, is considered a major risk factor associated with mental disorders, as shown in Figure 2.

Stress induces the activation of the sympathetic–adrenal–medullary and hypothalamic–pituitary–adrenal (HPA) systems, wherein the hypothalamus initiates the ‘fight or flight’ response. This response results in elevated heart rate, blood pressure, and glucose levels, suggesting a potential link between stress and mental health disorders due to the dysfunction of the HPA axis, which may lead to cellular changes [80,81,82,83,84].

There are several mechanisms through which stress is thought to influence mental status (Figure 3).

Stress can lead to the activation of brain immune cells, resulting in the release of pro-inflammatory mediators. This process may enhance the expression of inflammatory genes and promote lipid peroxidation, ultimately jeopardizing both the structure and function of neurons [85]. Chronic stress may affect brain pathology through with crucial mechanism such as tau hyperphosphorylation and synapse missorting [86]. Another possible link is oxidative stress, in large due to the production of reactive oxygen species (ROS) causing an inefficient mitochondrial function [87]. Furthermore, stress may cause an impairment of neuronal functions and structures such as the signaling pathway Fyn kinase and the NR2B-containing N-methyl-D-aspartate (NMDA) receptors, causing a reduction in GABA-ergic neurons and a decrease in Reelin protein expression [88,89].

As a natural response of the body when presented with adverse, demanding, or challenging conditions, stress causes a strain on both physical and mental functions. While the body induces adaptive responses designed to combat this strain, prolonged or excessive stress can lead to depression, anxiety, and cognitive dysfunction and may even increase the risk of development of mental and physical disease [83,90,91].

Chronic stress is known to lead to cognitive impairment. This is most likely due to elevated levels of glucocorticoids and corticotropin-releasing hormone causing neurotransmitter alterations, atrophy in the hippocampus and cortex, and shortening of the dendritic branches [92,93,94,95,96,97,98,99,100].

Stress is frequently involved in development of anxiety and depression. Affecting approximately 300 million people worldwide, anxiety and depression are one of the most prevalent and debilitating mental disorders [101,102,103]. Persistent systemic inflammation indicators, such as elevated pro-inflammatory cytokines, myelopoiesis and lymphopoiesis irregularities, and gut–epithelium and blood–brain barrier disturbances, can be found in individuals with stress-induced pathological conditions [104,105,106,107,108].

There is a significant interrelationship among depression, anxiety, and cognitive impairments, as well as cognitive decline. Anxiety has been linked to an elevated risk of both the onset and progression of somatic diseases [109,110]. Furthermore, a correlation has been identified between this psychiatric condition and complications arising from cardiovascular diseases [110]. Individuals suffering from insulin resistance who also experience anxiety are at a heightened risk of developing type 2 diabetes, with the coexistence of these two conditions further increasing the probability of adverse outcomes [111,112]. Each of the aforementioned disorders is linked to a greater likelihood of developing additional mental health issues [109,113]. Specifically, anxiety is associated with a higher risk of developing coexisting anxiety disorders or depression [109]. Depression itself is further connected to neurotic, somatoform, and personality disorders, as well as substance abuse and behavioral syndromes. The prevalence of comorbidity tends to escalate with the severity of depressive symptoms [113,114,115,116,117]. Moreover, depression and anxiety are closely associated with cognitive impairments and dementia. Although AD mainly influences cognitive abilities and memory, neuropsychiatric symptoms often emerge during the progression of the disease. The anxiety prevalence ranges from 9.4% during the preclinical phase to as high as 39% in cases of mild to severe cognitive decline, whereas the depression prevalence varies between 14.8% and 40% in mild to moderate stages of Alzheimer’s disease [118,119,120,121,122,123,124].

Mild cognitive impairment (MCI) is a significant transitional stage between typical aging and the onset of dementia. The prevalence of MCI among the elderly population is estimated to range from 5.0% to 36.7%, with approximately 11% to 13% of individuals with MCI progressing to dementia within a two-year period [125,126,127,128]. Psychological symptoms frequently associated with MCI include depression and anxiety, both of which can adversely affect cognitive functioning and contribute to the progression toward dementia [129,130]. Notably, cognitive deficits are more pronounced in individuals experiencing both depression and anxiety [131,132]. Depression is recognized as a major risk factor contributing to the progression of cognitive decline and the onset of dementia, whereas anxiety may affect this progression in both direct and indirect ways [133,134,135,136].

Among individuals with MCI, the prevalence of depression is approximately 32%. The use of antidepressants does not appear to provide a protective effect against the onset of dementia, and those with MCI are at an increased risk of developing severe cognitive impairment [137,138,139]. The conversion rate from MCI to dementia is notably higher in those with depression, ranging from 25% to 28%, with a significant conversion rate of 31% to AD in depressed patients compared to 13.5% in those without depression [140]. While anxiety has not been researched as thoroughly as depression, its connection to cognitive function is intricate, with prevalence rates ranging from 9.9% to 52% among individuals with MCI, especially affecting executive functions [134,141,142,143,144]. Dementia and anxiety can serve as significant clinical indicators; specifically, anxiety may act as a predictor of cognitive deterioration, whereas depression can help identify individuals with MCI who are at an increased risk of progressing to Alzheimer’s disease [140,144].

### 3.3. CNS-Activity of Satureja montana

According to data presented by the World Health Organization, roughly 4% of the worldwide population experienced anxiety disorders as of 2019 [145]. Moreover, it is important to acknowledge that a considerable number of anxiety disorders frequently remain undiagnosed [146]. Various obstacles are linked to the treatment of depression and anxiety, including variables such as age, polypharmacy or polypragmasy, an escalation in adverse drug reactions, comorbid conditions, a decline in quality of life, and increasing treatment expenditures. The COVID-19 pandemic and the associated restrictions have resulted in an over 25% rise in the incidence of these psychiatric disorders [147]. All pharmacological agents prescribed for the management of depressive disorders and anxiety are associated with possible adverse effects, including addiction, headaches, seizures, sexual dysfunction, suicidal ideation, and dependence [148]. Given the potential for adverse effects and delayed therapeutic outcomes, numerous patients exhibit hesitance regarding the prescription of psychotropic medications, which may lead to non-adherence to the suggested treatment [149,150]. The constraints associated with the utilization of these medications highlight the imperative for research into novel therapeutic alternatives for these conditions [148,151].

Ethnopharmacology reports the potential of SM to affect some stress-related chronic mental diseases, including anxiety, depression, mild cognitive impairments, and dementia. In a rat model designed to assess acute stress, the dry extract of SM significantly enhanced locomotor activity and extended the social interaction duration while concurrently diminishing anxiety-related behaviors [79]. Acute cold stress substantially reduced the duration of novel-object-recognition time in laboratory rats, indicating a significant anxiolytic effect associated with SM. The implications of neurotransmitters, specifically 5-hydroxytryptamine and gamma-aminobutyric acid (GABA), were examined [152]. The bioactive constituents in *Satureja montana*, including rosmarinic acid and carvacrol, exhibited moderate anxiolytic effects [79]. In a rat model subjected to chronic stress, the dry extract of SM revealed anxiolytic properties. The spontaneous locomotor activity of the experimental rats was also altered [153,154], reinforcing certain behavioral assessments aimed at evaluating both anxiolytic and antidepressant effects [152,155]. Research has established that rosmarinic acid affects T-type calcium ion channels within the central nervous system, in addition to influencing GABA-ergic and cholinergic pathways [71]. A study from 2022 proposed that the anxiolytic mechanism may primarily involve the modulation of ion channels within the central nervous system and cholinergic mediation, along with various other mechanisms contributing to the anxiolytic properties of the dry extract of SM [79]. Furthermore, the investigation carried out by Melo et al. (2010) suggests that carvacrol interacts with a range of neurotransmitters, including GABA, noradrenaline, and serotonin [156]. In addition to the findings by Vilmosh et al. (2022) [79], our comprehensive literature review did not reveal any studies that specifically examine the effects of the dry extract of SM on anxiety. It was concluded that the dry extract may exhibit a more pronounced anxiolytic effect in comparison to the individual active compounds when utilized in isolation. This observation supports the hypothesis of a synergistic interaction between the principal components, rosmarinic acid and carvacrol [79].

Chronic low-grade inflammation and oxidative stress are implicated in the etiology of anxiety, which correlates with elevated plasma levels of corticosterone, a byproduct of stress responses [157,158,159]. Consequently, when evaluating the anxiolytic properties of rosmarinic acid and carvacrol, it is essential to consider their antioxidant and anti-inflammatory effects [72,160,161].

Epidemiological studies indicate that the prevalence of depression aligns closely with that of anxiety disorders. The presence of depressive disorder significantly diminishes an individual’s quality of life [162]. This condition typically manifests in an episodic manner; however, it remains unpredictable regarding the frequency and duration of episodes, as well as the efficacy of various treatment options. The recovery process is often extended, accompanied by a considerable risk of relapse [163]. Similar to anxiety, depression acts as a distinct risk factor for various health complications, encompassing diabetes, cardiovascular ailments, chronic respiratory diseases, and arthritis, in addition to cognitive deficits and neurodegenerative conditions like AD.

An examination of the effects of *Satureja montana* and its principal constituents, specifically rosmarinic acid and carvacrol, on recognition memory, social interaction, and depressive behaviors was conducted in experimental models [79]. The dry extract of SM demonstrated notable enhancements in the exploration duration of novel objects and the length of social interactions, particularly at dosages of 250 mg/kg and 500 mg/kg body weight. The higher dosage significantly influenced the discrimination index (DI), indicating an improvement in recognition memory. In contrast, while rosmarinic acid exhibited some anxiolytic properties, it did not exert a significant effect on social interaction or recognition memory; similarly, carvacrol did not demonstrate any meaningful effects on these parameters. The extract of *Satureja montana* displayed more pronounced anxiolytic and antidepressant effects compared to its individual components, likely owing to synergistic interactions among its constituents. The findings regarding rosmarinic acid are consistent with the existing literature, which underscores the anxiolytic properties of this phenolic compound [164,165,166]. In a manner analogous to rosmarinic acid, carvacrol has also been reported to enhance cognitive function in rodent models [167,168], which contrasts with the findings presented by Vilmosh et al. (2022) [79]. These discrepancies may be elucidated by the variations in the memory impairment models utilized and the differing treatment durations with the phenolic compounds.

The study revealed that the dry extract of *Satureja montana* significantly augmented the exploration time of a novel object in recognition memory assessments, with the higher dosage of 500 mg/kg body weight. Rosmarinic acid also manifested a significant impact on recognition memory, whereas carvacrol did not exhibit any notable influence. Nevertheless, the existing literature presents contradictory findings regarding the effects of rosmarinic acid on recognition memory [164,165].

The dry extract of *Satureja montana* exhibited robust anxiolytic and antidepressant effects, surpassing both rosmarinic acid and carvacrol when administered separately [79]. In social interaction assessments, the dry extract of SM improved the duration of interactions in rats subjected to chronic stress, whereas rosmarinic acid did not produce similar outcomes. The findings imply that the synergistic effects of the compounds within the dry extract may contribute to its enhanced cognitive and emotional benefits compared to the individual components. According to two separate studies conducted in 2021, rosmarinic acid has been documented to yield a significant antidepressant effect by decreasing the duration of immobility in the forced swim test [169,170]. Conversely, Vilmosh et al. (2022) noted that rosmarinic acid influenced depressive-like behavior solely within the framework of a chronic stress model [79]. This discrepancy may stem from the prolonged treatment duration and the distinct physiological mechanisms activated in the two stress models employed. Chronic stress is associated with a decline in the activity of dopaminergic neurons in the ventral tegmental area, while rosmarinic acid is recognized for its ability to elevate dopamine levels, thereby clarifying its observed antidepressant effects [171,172].

In contrast to the findings related to rosmarinic acid, the investigation conducted by Vilmosh et al. (2022) revealed that carvacrol did not exhibit any antidepressant properties [79], thus calling into question the interpretations established within the existing literature [173,174]. This divergence may stem from differences in the dosages administered and the methodologies utilized to induce depressive states. Although carvacrol is recognized for its effects on dopamine and serotonin neurotransmission [134], it is conceivable that these pathways are inadequate to mitigate the stress-induced disruptions within the central nervous system.

### 3.4. Neuroprotective Mechanisms of Satureja montana

#### 3.4.1. Antioxidant Activity

*Satureja montana* contains a wealth of phenolic compounds, including rosmarinic acid, caffeic acid, and flavonoids such as rutin. These compounds are recognized for their potent antioxidant capabilities, which can alleviate oxidative stress, a critical contributor to neurodegenerative disorders. Furthermore, the essential oils derived from SM have shown considerable radical scavenging activity, suggesting their ability to counteract free radicals and diminish oxidative harm.

The majority of the advantageous effects of Balkan savory on human health can be attributed to its notable antioxidant and anti-inflammatory properties. Research has confirmed the presence of free radical-scavenging activity through three distinct in vitro methods, leading the authors to classify the examined dry extract of SM as an effective antioxidant. Notably, SM exhibited superior antioxidant activity when compared to findings associated with other species within *Satureja* sp. [34]. Both in vitro and in vivo studies demonstrated SM’s ability to neutralize free radicals and to influence the endogenous antioxidant system of the human body. Simultaneously, it was determined that SM possesses a local anti-inflammatory effect that is comparable to that of diclofenac. SM demonstrates a propensity to lower serum levels of IL-6, IL-1b, and TNF in both acute and chronic stress models. The authors proposed that the inhibition of cyclooxygenase is the most probable mechanism underlying the action of SM. In addition, akin to its antioxidant properties, the anti-inflammatory effect of the extract is notably more significant when compared to the effects of rosmarinic acid and carvacrol used independently [34].

The neuroprotective antioxidant mechanisms of compounds obtained from *Satureja montana* encompass various biochemical pathways and functions, as illustrated in Table 3.

Direct Antioxidant Activity. Extracts of *Satureja montana* demonstrate considerable antioxidant properties in vitro, primarily attributed to their ability to scavenge free radicals, thereby mitigating oxidative stress. An in vitro study conducted in 2007 highlighted the potential of *Satureja montana* L. subsp. kitaibelii extract as a natural source of antioxidants and antimicrobial agents [175]. The research indicated that phenolic compounds present in plant extracts demonstrate considerable antioxidant activity by neutralizing free radicals and inhibiting the breakdown of hydroperoxides. The total phenolic content was assessed using the Folin–Ciocalteu method and HPLC, revealing strong correlations between total phenolic content and antioxidant activity across various assays, with r2 values ranging from 0.62 to 0.90.

**Table 3 cimb-47-00556-t003:** Mechanisms of antioxidant activity of *Satureja montana*-derived compounds.

Mechanism	Description	Ref.
Direct Antioxidant Activity	Scavenging of free radicals, reducing oxidative stress	[176]
Enzyme Modulation	Increased activity of SOD, CAT, and GSR	[35,177]
Reduction in Lipid Peroxidation	Decreased lipid peroxidation, protecting cell membranes	[33,177]
Anti-inflammatory Effects	Decreased levels of TNF-α and IL-6	[176]
Hepatoprotective Effects	Reduced oxidative stress enzymes and inflammatory cells in the liver	[33]
Neuroprotective Pathways	Activation of Nrf2/ARE pathway, enhancing antioxidant protein expression	[178,179]
Drug Delivery Potential	Use of niosomes for stable and targeted delivery of antioxidants	[180]

Furthermore, the antioxidant activity of SM varietettes was assessed utilizing three methods, revealing that the essential oils exhibited remarkable radical scavenging effects against the ABTS radical, with an IC50 ranging from 30.02 to 34.5 µg/mL [65].

The antioxidant characteristics, as part of the anti-inflammatory effects of Balkan savory extract, were reported in 2024 [176]. The antioxidant properties of the dry SM extract were validated using three distinct in vitro techniques; however, the activity observed was lower than what has been documented in prior research, possibly attributable to differences in the extract compositions.

Enzyme Modulation. The SM extract has demonstrated an enhancement in the activity of essential antioxidant enzymes, including superoxide dismutase (SOD), catalase (CAT), and glutathione reductase (GSR) [33,177]. These enzymes are vital for the neutralization of ROS and the preservation of cellular redox equilibrium.

Reduction in Lipid Peroxidation. Extracts of *Satureja montana* diminish lipid peroxidation, a significant indicator of oxidative damage to cellular membranes. This mitigation plays a crucial role in safeguarding neuronal cells against oxidative injury [33,177].

Anti-inflammatory Effects. SM extracts demonstrate anti-inflammatory effects by reducing the concentrations of pro-inflammatory cytokines, including TNF-α and IL-6, in specific stress models [176]. This is significant because inflammation frequently occurs alongside oxidative stress in neurodegenerative disorders.

Hepatoprotective Effects through Antioxidant Activity. In studies utilizing animal models, extracts of *Satureja montana* have demonstrated hepatoprotective properties by diminishing oxidative stress enzymes and inflammatory cell presence in the liver, thereby indirectly bolstering the overall antioxidant defense systems [33].

Neuroprotective Pathways. While research specifically focusing on *Satureja montana* and its neuroprotective effects is scarce, analogous compounds derived from other botanical sources have been shown to stimulate the Nrf2/ARE signaling pathway, thereby increasing the production of antioxidant proteins and offering neuroprotective benefits [178,179]. This pathway plays a vital role in protecting cells from oxidative damage.

Potential for Drug Delivery. Novel methodologies, including the utilization of niosomes infused with *Satureja* extracts, have demonstrated potential in augmenting the stability and precise delivery of antioxidants to neural tissues, consequently enhancing neuroprotective effects [180].

#### 3.4.2. Anti-Inflammatory Activity

The phenolic compounds found in *Satureja montana*, such as rosmarinic acid, exhibit anti-inflammatory properties. Chronic inflammation plays a significant role in neurodegeneration, and mitigating this inflammation may safeguard neuronal cells. The anti-inflammatory properties of compounds derived from SM, especially concerning neuroprotection, involve multiple biochemical pathways. The following outlines the principal mechanisms identified in the current literature:

Oxidative Stress Reduction. The dry extract of SM demonstrates considerable antioxidant properties, which play a vital role in alleviating oxidative stress, a prevalent factor in neuroinflammation and neurodegeneration. This antioxidant capability aids in neutralizing free radicals and diminishing ROS, thus safeguarding neuronal cells from oxidative harm [176].

Modulation of Pro-inflammatory Cytokines. In a model of acute stress, the dry extract of *Satureja montana* administered at a dosage of 250 mg/kg markedly reduced the concentrations of pro-inflammatory cytokines, including TNF-α and IL-6, when compared to other substances such as carvacrol and rosmarinic acid [176]. This decrease in cytokine levels suggests its potential role in regulating inflammatory responses within the brain.

Blood–Brain Barrier (BBB) Protection. Carvacrol, a principal constituent of SM, has demonstrated efficacy in reducing brain edema and inhibiting blood–brain barrier permeability following traumatic brain injury. This effect is mediated by its antioxidant properties, which lower levels of malondialdehyde and ROS while enhancing SOD activity and overall antioxidant capacity [181]. Preserving the integrity of the BBB is crucial for averting neuroinflammation and the resulting neuronal injury.

Matrix Metalloproteinase Inhibition. Carvacrol administration additionally inhibits the expression of matrix metalloproteinase-9 (MMP-9), an enzyme responsible for the degradation of extracellular matrix components and compromising blood–brain barrier (BBB) integrity. Through the inhibition of MMP-9, carvacrol contributes to the maintenance of the structural integrity of the BBB [181].

Neuroprotective Pathways. Natural compounds, such as those derived from *Satureja montana*, exhibit neuroprotective properties primarily through the activation of signaling pathways, including the phosphoinositide 3-kinase (PI3K)/Akt and mitogen-activated protein kinase (MAPK) pathways. These pathways are integral to antioxidant defense mechanisms and cellular survival, thereby enhancing neuroprotection [182].

The neuroprotective anti-inflammatory mechanisms of Satureja montana encompass various biochemical pathways and functions, as illustrated in Table 4.

#### 3.4.3. Anti-Apoptotic Properties

The anti-apoptotic properties of Balakan savory, particularly in relation to neuroprotection, can be deduced from its observed effects in other biological systems, given the scarcity of direct studies on its neuroprotective mechanisms. Research on a rat model indicates that *Satureja montana* extract reduces the expression of pro-apoptotic genes such as Fas and Bax while enhancing the expression of anti-apoptotic genes like Bcl-2 in testicular tissue, thereby suggesting a potential mechanism for neuronal protection against apoptosis [183]. Furthermore, SM extract exhibits notable antioxidant activity, effectively reducing lipid peroxidation and restoring glutathione levels, which are vital for safeguarding cells from oxidative stress-related apoptosis. The extract also decreases pro-apoptotic proteins like Fas and Bax while increasing anti-apoptotic proteins such as Bcl-2, establishing a critical balance necessary for preventing cell death. Additionally, *Satureja montana* extract enhances the expression of peroxisome proliferator-activated receptor-gamma (PPAR-γ) and normalizes Akt1 protein levels, both of which play significant roles in cell survival pathways, promoting growth and inhibiting apoptosis. This study further demonstrated that the extract mitigates DNA fragmentation, a key indicator of apoptosis, underscoring its importance in maintaining cellular integrity, neuronal health, and function.

#### 3.4.4. Acetylcholinesterase Inhibition

Inhibition of acetylcholinesterase (AChE) can enhance cholinergic transmission, which is beneficial in conditions like Alzheimer’s disease where cholinergic deficits are prominent. In a comparative analysis of different species within the Lamiaceae family, *Satureja montana* emerged as one of the plants demonstrating significant acetylcholinesterase inhibitory activity at a concentration of 1 mg/mL [37]. Additionally, essential oils from *Satureja montana* inhibited human serum cholinesterase activity, indicating the potential for treating neurological diseases [184]. Further, extracts from SM, obtained through supercritical fluid extraction, showed significant inhibition of butyrylcholinesterase (BChE). The nonvolatile fractions, rich in bioactive compounds including (+)-catechin, chlorogenic acid, vanillic acid, and protocatechuic acid, were particularly effective [38]. The main active compounds identified in *Satureja* species include carvacrol, γ-terpinene, thymol, and rosmarinic acid, which contribute to their cholinesterase inhibitory activity [185,186]. Essential oils extracted from SM also demonstrated inhibitory effects on AChE, suggesting their potential use in therapeutic applications [39,40].

Mechanisms of neuroprotective activity of *Satureja montana* are summarized in Figure 4.

## 4. Discussion

### 4.1. Neuroprotective and Psychotropic Potential of Satureja montana

This review emphasizes the significant neuropharmacological capabilities of *Satureja montana*, a plant abundant in bioactive substances like carvacrol and rosmarinic acid. The dry extract of SM exhibited enhanced anxiolytic, antidepressant, and cognitive-enhancing properties in both acute and chronic stress models when compared to its individual components. These results indicate a synergistic interaction among its phytochemicals, highlighting the therapeutic advantages of utilizing whole-plant extracts over the administration of isolated compounds.

### 4.2. Mechanistic Insights into CNS Activity of Satureja montana

The anxiolytic and antidepressant properties of SM are believed to arise from its modulation of central neurotransmitters, especially serotonin, dopamine, and noradrenaline. Furthermore, its engagement with GABA-ergic and cholinergic pathways, along with its effect on T-type calcium channels, enhances its neuroactive characteristics. The extract has also been shown to alleviate stress-induced behavioral impairments in experimental models, which include changes in locomotor activity, social interaction, and immobility duration in the Forced Swim test. These behavioral results align with mechanisms associated with emotional regulation and neurochemical equilibrium.

Oxidative stress and chronic inflammation play pivotal roles in the development of neurodegenerative and psychiatric disorders. SM demonstrates significant antioxidant capabilities, boosting the function of natural antioxidant enzymes such as superoxide dismutase, catalase, and glutathione reductase. It effectively diminishes lipid peroxidation and neutralizes ROS, thus safeguarding neuronal health. Simultaneously, its anti-inflammatory properties are facilitated by the downregulation of pro-inflammatory cytokines (such as TNF-α and IL-6) and the inhibition of cyclooxygenase activity, exhibiting a potency that surpasses that of its individual components.

The extract enhanced recognition memory and the ability to discriminate novel objects, particularly in animal models experiencing chronic stress. The cognitive benefits were not as pronounced when rosmarinic acid or carvacrol was used separately, suggesting that a synergistic effect is observed in the full extract. Notably, SM demonstrated considerable inhibitory activity against AChE, highlighting its potential for cholinergic enhancement, which is a crucial focus in the treatment of Alzheimer’s disease.

In addition to its antioxidant and anti-inflammatory properties, SM also affects anti-apoptotic signaling. It reduces the expression of pro-apoptotic genes such as Fas and Bax while enhancing the levels of anti-apoptotic markers like Bcl-2. Furthermore, it modulates survival pathways including Akt and PPAR-γ, which are essential for preventing neuronal death and maintaining cognitive function, especially in the context of chronic stress.

### 4.3. Clinical Implications and Future Directions

While the findings are encouraging, the discourse highlights significant deficiencies in the existing body of literature. There is a lack of clinical research assessing the effectiveness and safety of *Satureja montana* in human subjects. The majority of investigations have focused on animal studies or in vitro experiments, which, although valuable, require careful consideration prior to application in clinical settings. In vitro and animal research frequently employs concentrations and methods of administration that are challenging to replicate in human subjects. Variations in metabolism, permeability of the blood–brain barrier, and systemic bioavailability of these substances present considerable obstacles. Additionally, differences in experimental designs, treatment lengths, and dosages of compounds lead to variations in outcomes, especially when analyzing the distinct impacts of rosmarinic acid and carvacrol. The lack of standardized extract formulations and discrepancies in study design further complicate the ability to compare findings across different studies, thereby diminishing their clinical relevance. This review primarily examines the mechanistic properties of the neuroprotective effects attributed to the key components found in savory extracts. However, forthcoming clinical studies that concentrate on optimal dosing strategies, suitable pharmaceutical formulations, and specific target populations will be crucial for their implementation in therapeutic settings.

## 5. Conclusions

*Satureja montana* may represent a valuable and promising alternative in the prevention and treatment of various mental health disorders, including dementia. The main mechanisms by which it affects pathological processes in the CNS are significant anxiolytic activity, anticholinergic activity, and strong antioxidant and anti-inflammatory properties producing moderate cognitive benefits. *Satureja montana* exhibits several properties that could contribute to neuroprotection, including antioxidant, anti-inflammatory, and anti-apoptotic effects, modulation of neurotrophic factors, and acetylcholinesterase inhibition. These mechanisms collectively suggest that Balkan savory has the potential to protect against neurodegenerative diseases and support overall neuronal health. However, despite there being existing literature regarding the composition and biological activity of *Satureja montana*, further research is essential to elucidate its established pharmacological effects and their potential complex mechanisms.

## Figures and Tables

**Figure 1 cimb-47-00556-f001:**
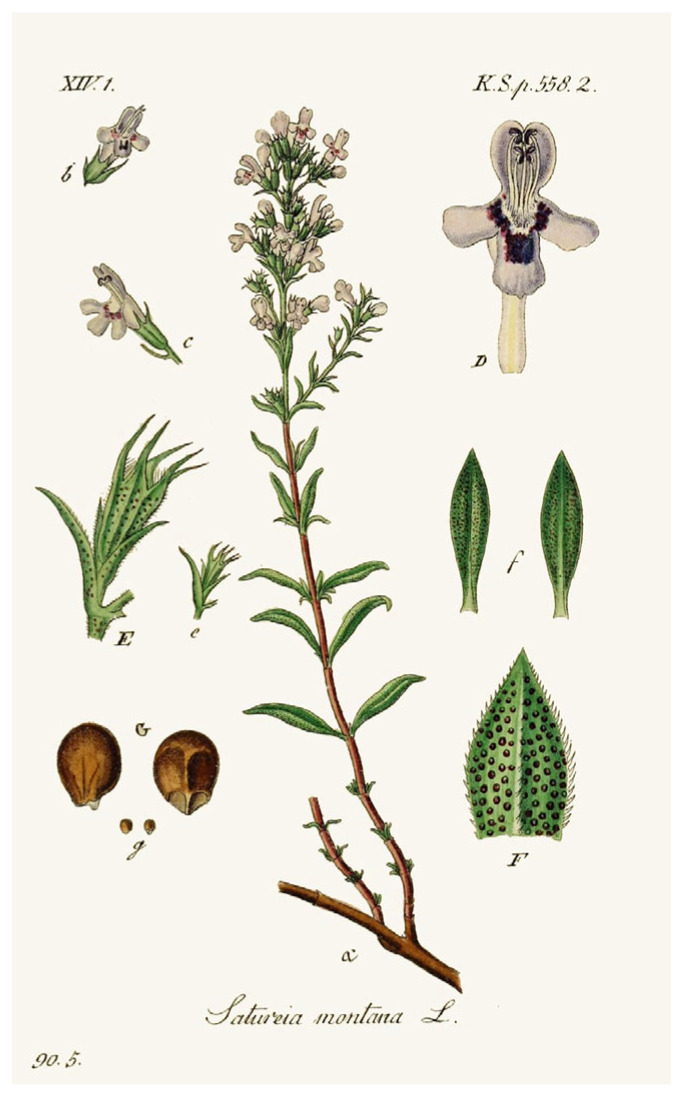
*Satureja montana* from Jacob Sturm, public domain, via Wikimedia Commons (https://commons.wikimedia.org/wiki/File:Satureja_montana_-_Deutschlands_flora_in_abbildungen_nach_der_natur_-_vol._20_-_t._17.jpg, accessed on 10 July 2025).

**Figure 2 cimb-47-00556-f002:**
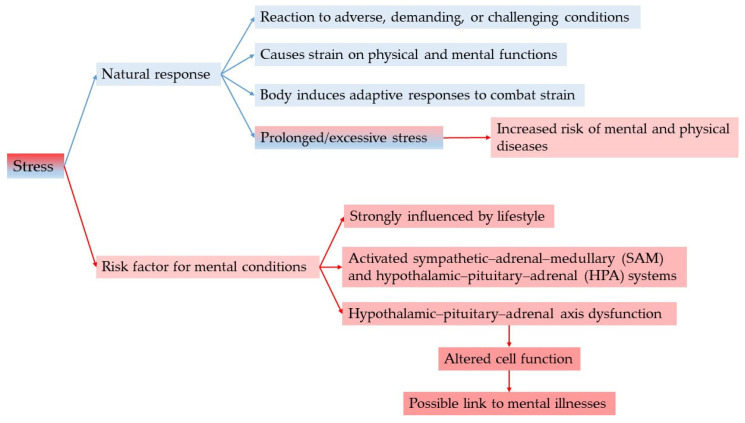
The dual role of stress.

**Figure 3 cimb-47-00556-f003:**
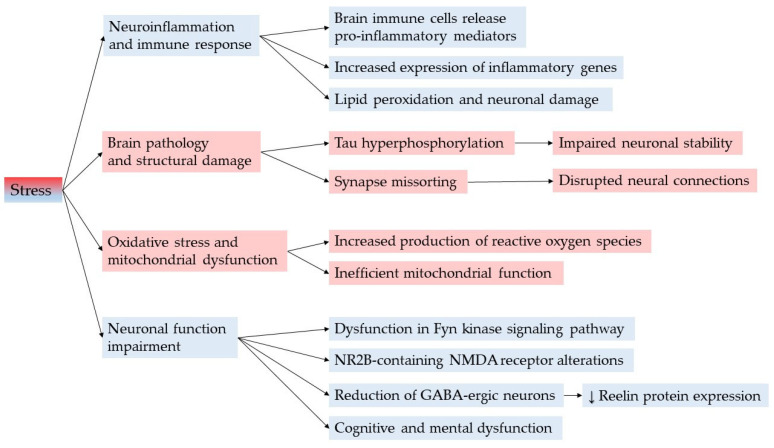
Molecular mechanisms linking stress to mental diseases.

**Figure 4 cimb-47-00556-f004:**
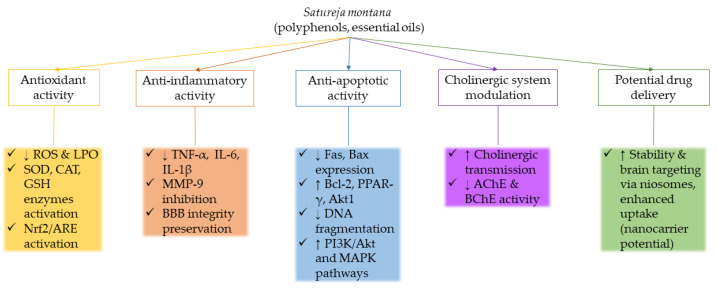
Summary of the neuroprotective mechanisms of *Satureja montana*.

**Table 1 cimb-47-00556-t001:** Main phenolic components of *Satureja montana* extracts.

Component	Class	Notes	Ref.
Carvacrol	Phenolic component	Phenolic monoterpene	[68,69]
Thymol	Phenolic component	Phenolic monoterpene	[26,64,65]
Rosmarinic acid	Phenolic component	Phenolic acid	[28,29,41]
Caffeic acid	Phenolic component	Phenolic acid	[26,28,70]
Chlorogenic acid	Phenolic component	Phenolic acid	[26,28,70]
Ellagic acid	Phenolic component	Phenolic acid	[26,28,70]
Quercetin	Flavonoid	Flavonol	[41]
Quercetin-3-O-α-L-rhamnopyranoside	Flavonoid	Quercetin glycoside	[41]
Quercetin-7-O-glucopyranoside	Flavonoid	Quercetin glycoside	[41]
Luteolin-7-rhamnoside-4′-O-β-glucopyranoside	Flavonoid	Luteolin derivative	[41]
Luteolin-7-O-glucopyranoside	Flavonoid	Luteolin derivative	[41]
Rutin	Flavonoid	Quercetin glycoside	[28]

**Table 2 cimb-47-00556-t002:** Content and biological properties of major phenolic components of *Satureja montana* extracts.

Component	Percentage/Presence	Activity	Ref.
Carvacrol	44.5–45.7%	Antimicrobial, Antioxidant	[64,65]
		Anxiolytic, Antidepressant	[71,72]
p-Cymene	12.6–16.9%	Antimicrobial	[64,65]
γ-Terpinene	8.1–8.7%	Antioxidant	[64,65]
Thymol	Up to 81.79%	Antimicrobial, Antioxidant	[64,65]
Rosmarinic Acid	Major phenolic compound	Antioxidant, Anti-inflammatory	[28,29,41,71]
		Anxiolytic, Antidepressant	[71,72,78]
Caffeic Acid	Present	Antioxidant	[28,70]
Chlorogenic Acid	Present	Antioxidant	[28,70]
Ellagic Acid	Present	Antioxidant, Anti-inflammatory	[28,70]
Quercetin	Present	Antioxidant, Antimicrobial	[41]
Luteolin	Present	Antioxidant	[41]
Rutin	Present	Antioxidant	[26,28]

**Table 4 cimb-47-00556-t004:** Mechanisms of anti-inflammatory activity of *Satureja montana*.

Mechanism	Description	Ref.
Antioxidant Activity	Scavenges free radicals and reduces ROS	[42,181]
Cytokine Modulation	Decreases pro-inflammatory cytokines (TNF-α, IL-6)	[42]
BBB Protection	Reduces brain edema and prevents BBB permeability	[181]
MMP-9 Inhibition	Suppresses MMP-9 expression, preserving BBB integrity	[181]
Signaling Pathways	Activates PI3K/Akt and MAPK pathways for cell survival	[182]

## Data Availability

No new data were created.

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
