# Peer review of "Protective Potential of Satureja montana-Derived Polyphenols in Stress-Related Central Nervous System Disorders, Including Dementia"

_cimb, 2025, doi:10.3390/cimb47070556_

Round 1

Reviewer 1 Report

Comments and Suggestions for Authors

It is a good review on preliminary studies directed to the use of extract of already used ethnopharmaceutical for possible curing of mental diseases. PAper is well organized, comprehensive and well written. Although number of such reviews grows enormously they reflect vigrous growth of studies on ethnopharmaceuticals, which usually act as synergetic mixtures of variable compounds. It is well pointed out also in this manuscriot. Therefore paper is ready to be published, however, with minute editorial corrections (they could be made upon proofreeding):

1./ all the latin names of Balkan savory should be in italics (see patagraph 3.1.);

2./ line 140; should be p-cymene;

3./ line 267: there should be indicated that this animal model were rats.

Author Response

Dear Reviewer 1,

Thank you for the high rating and encouragement.

All technical remarks have been corrected.

Reviewer 2 Report

Comments and Suggestions for Authors

Dear authors of the manuscript!

I have studied your research with great interest. Satureja Montana exhibits several properties that could contribute to neuroprotection, including antioxidant, anti-inflammatory, anti-apoptotic effects, modulation of neurotrophic factors, and acetylcholinesterase inhibition. These mechanisms collectively suggest that Balkan savory has the potential to protect against neurodegenerative diseases and support overall neuronal health. However, despite existing literature regarding the composition and biological activity of Satureja montana, further research is essential to elucidate its established pharmacological effects and their potential complex mechanisms.

However, it would be desirable to supplement your undoubtedly very interesting study with the following points:
1. The authors provide Table 1 "Main components of Satureja montana extracts" in the text. However, the data in the table is very, very limited. It is necessary to expand the main components in the article. Satureja montana has a very rich polyphenol content, it is enough to find literally any article on the mass spectrometric study of this plant. And as is known from the latest research, it is not each specific individual chemical compound that produces a positive effect, but rather their synergistic effect. This should also be noted.
2. Accordingly, the table 2 "Content and biological properties of major phenolic components of S. montana" is very limited. The positive pharmaceutical effect of the flavones group and flavonols group is not provided.
3. It is also desirable to include in the text the methods of extraction of S. montana to obtain medicinal infusions. What methods are currently recognized as the most effective for the plant in the development of the pharmacopoeia?

Author Response

Reviewer 2

“Dear authors of the manuscript!

I have studied your research with great interest. Satureja Montana exhibits several properties that could contribute to neuroprotection, including antioxidant, anti-inflammatory, anti-apoptotic effects, modulation of neurotrophic factors, and acetylcholinesterase inhibition. These mechanisms collectively suggest that Balkan savory has the potential to protect against neurodegenerative diseases and support overall neuronal health. However, despite existing literature regarding the composition and biological activity of Satureja montana, further research is essential to elucidate its established pharmacological effects and their potential complex mechanisms.

However, it would be desirable to supplement your undoubtedly very interesting study with the following points:
1. The authors provide Table 1 "Main components of Satureja montana extracts" in the text. However, the data in the table is very, very limited. It is necessary to expand the main components in the article. Satureja montana has a very rich polyphenol content, it is enough to find literally any article on the mass spectrometric study of this plant. And as is known from the latest research, it is not each specific individual chemical compound that produces a positive effect, but rather their synergistic effect. This should also be noted.

Answer: We appreciate the reviewer’s comment regarding the limited scope of Table 1. Indeed, like many medicinal plants, Satureja montana contains a wide array of biologically active compounds, particularly diverse polyphenols. However, in the present review, our aim is to focus on the main phenolic constituents that are most frequently reported in the literature and typically present in the highest concentrations. To avoid any ambiguity, we have revised the title of Table 1 to clearly indicate that it includes only the principal phenolic compounds identified in Satureja montana extracts. Additionally, the text discusses the synergistic effect of active principles obtained from the plant extracts, as illustrated in lines 292-293, 317-319, 336-338, and discussed in lines 503-505, and 524-526.

  1. 2. Accordingly, the table 2 "Content and biological properties of major phenolic components of S. montana" is very limited. The positive pharmaceutical effect of the flavones group and flavonols group is not provided.

Answer: We thank the reviewer for pointing out this important omission. In response, we have revised Table 2 to include additional data on the positive pharmaceutical effects of flavones and flavonols present in Satureja montana, mentioned in Table 1.

  1. 3. It is also desirable to include in the text the methods of extraction of S. montana to obtain medicinal infusions. What methods are currently recognized as the most effective for the plant in the development of the pharmacopoeia?”

Answer: We are grateful to the reviewer for the suggestion. This issue is quite extensive and debatable and, in our opinion, deserves a separate consideration in a separate article, which could be the focus of our future research.

We express our gratitude to reviewer 2 for his interest and the time he dedicated to this project. We believe the suggestions provided and the correction made contribute in enhancing the quality of our review.

Reviewer 3 Report

Comments and Suggestions for Authors

The manuscript focuses on the neuroprotective potential of polyphenols derived from Satureja montana in stress-related mental disorders, including dementia. The topic is highly relevant, as it integrates ethnopharmacological research with modern approaches to phytotherapeutic interventions in neurodegenerative diseases. Within the global trend of re-evaluating plant-based treatments for central nervous system (CNS) disorders, this work is both timely and scientifically justified.

The objectives are clearly formulated. The manuscript is well-structured, with distinct sections: introduction, methodology, results, discussion, and conclusions. Tables and figures are useful and facilitate the understanding of data and proposed mechanisms. The literature review is robust and covers recent studies, particularly regarding rosmarinic acid and carvacrol. However, further differentiation between in vitro, in vivo, and clinical evidence would enhance the critical appraisal.

  1. The review lacks specific details regarding the inclusion/exclusion criteria and the quality assessment procedure of the selected studies. I strongly recommend completing the "Materials and Methods" section with explicit information on the screening strategy and evaluation of the literature sources.
  2. The authors present a valuable synthesis of S. montana phenolic compounds (notably rosmarinic acid and carvacrol) and discuss relevant mechanisms (antioxidant, anti-inflammatory, anxiolytic, anticholinesterase, anti-apoptotic). However, while the absence of clinical trials is acknowledged in the discussion, a more critical perspective on the translatability of preclinical findings to human applications is warranted.
  3. Certain paragraphs are overly dense and would benefit from syntactic simplification. The manuscript would also benefit from uniform exposition of data from the literature, to ensure better readability and comprehension.
  4. Minor typographical errors (e.g., "of of" in the abstract) were identified. A final proofreading is recommended to eliminate such inconsistencies.
  5. Consider adding an integrative figure summarizing all discussed neuroprotective mechanisms of Satureja montana, for better conceptual consolidation.
  6. The manuscript would benefit from a dedicated paragraph proposing clear future directions for clinical research (e.g., dosing schemes, formulation types, target populations, potential adverse effects).
  7. Currently, Figure 1 is not very informative—only the inflorescence is shown. I suggest replacing or supplementing it with a botanical illustration of the whole plant, including stem, leaves, roots, and seeds/fruits, to provide a more complete visual reference.
  8. More detailed data on the native and current distribution of S. montana would be valuable. Please specify whether the species is considered invasive in certain ecosystems.
  9. The terms "Fidelity Level (FL)" and "Ethnobotanicity Index (EI)" are used only once, and abbreviations are not reused throughout the text. Please choose to either (a) consistently use the full terms or (b) define and use the abbreviations uniformly. Ex.: the same applies to "SM" (Satureja montana), which is inconsistently abbreviated.
  10. All Latin names of species should be written in italic. However, taxonomic families (e.g., Lamiaceae) should remain in regular font.
  11. Please add a new column to Table 1 listing the references that reported the presence of the respective compounds.
  12. Consider adding bibliographic references in the same row with the described biological activities for improved traceability.
  13. Tables 2, 3, and 4 cite sources but use two different formatting styles. I recommend unifying the style of reference citation across all tables.

Author Response

Answers to Reviewer 3

The manuscript focuses on the neuroprotective potential of polyphenols derived from Satureja montana in stress-related mental disorders, including dementia. The topic is highly relevant, as it integrates ethnopharmacological research with modern approaches to phytotherapeutic interventions in neurodegenerative diseases. Within the global trend of re-evaluating plant-based treatments for central nervous system (CNS) disorders, this work is both timely and scientifically justified.

The objectives are clearly formulated. The manuscript is well-structured, with distinct sections: introduction, methodology, results, discussion, and conclusions. Tables and figures are useful and facilitate the understanding of data and proposed mechanisms. The literature review is robust and covers recent studies, particularly regarding rosmarinic acid and carvacrol. However, further differentiation between in vitro, in vivo, and clinical evidence would enhance the critical appraisal.

  1. The review lacks specific details regarding the inclusion/exclusion criteria and the quality assessment procedure of the selected studies. I strongly recommend completing the "Materials and Methods" section with explicit information on the screening strategy and evaluation of the literature sources.

Answer: We thank the reviewer for the valuable comment. In response, we have revised the “Materials and Methods” section and included a detailed description of the inclusion and exclusion criteria as well as the approach used to assess the methodological quality of the selected studies.

  1. The authors present a valuable synthesis of S. montana phenolic compounds (notably rosmarinic acid and carvacrol) and discuss relevant mechanisms (antioxidant, anti-inflammatory, anxiolytic, anticholinesterase, anti-apoptotic). However, while the absence of clinical trials is acknowledged in the discussion, a more critical perspective on the translatability of preclinical findings to human applications is warranted.

Answer: We thank the reviewer for this insightful comment. We fully agree that highlighting the limitations in translating preclinical evidence to human applications is essential. In response, we have revised the Discussion section to include a more critical reflection on the translatability of the reviewed findings (Section 4.3. Clinical Implications and Future Directions), emphasizing the need for well-designed clinical trials and discussing the potential pitfalls and challenges associated with extrapolating preclinical data to human health outcomes.

  1. Certain paragraphs are overly dense and would benefit from syntactic simplification. The manuscript would also benefit from uniform exposition of data from the literature, to ensure better readability and comprehension.

Answer: The reviewer's recommendation to simplify the language and standardize the presentation of information is extremely useful for improving the quality of the article. We have reworked the entire text in unison with this note.

  1. Minor typographical errors (e.g., "of of" in the abstract) were identified. A final proofreading is recommended to eliminate such inconsistencies.

Answer: We thank the reviewer for this recommendation. We carefully reviewed the entire text and removed the discovered errors.

  1. Consider adding an integrative figure summarizing all discussed neuroprotective mechanisms of Satureja montana, for better conceptual consolidation.

Answer: We appreciate the reviewer’s suggestion to include a visual summary of the discussed neuroprotective mechanisms. In response, we have added an integrative figure that illustrates the key pathways and biological effects through which Satureja montana and its bioactive compounds may exert neuroprotective effects replacing Table 5. We believe that this schematic serves to conceptually consolidate the findings presented in the review and improve overall clarity for the reader.

  1. The manuscript would benefit from a dedicated paragraph proposing clear future directions for clinical research (e.g., dosing schemes, formulation types, target populations, potential adverse effects).

Answer: We appreciate the reviewer’s insightful suggestion regarding the inclusion of a paragraph outlining clear future directions for clinical research (e.g., dosing schemes, formulation types, target populations, potential adverse effects). Indeed, this is a valuable and important aspect that deserves thorough investigation. However, we believe that such a discussion would require extensive elaboration and could form the basis of a dedicated future article. In the present review, our primary aim is to provide a comprehensive synthesis of the currently available data on the established mechanisms of neuroprotective action of key constituents in savory extracts. We have thus focused on mechanistic insights from mainly preclinical studies as a foundation for future translational efforts. To reflect the reviewer's recommendation without shifting the focus from the main objective, we added a brief addition in this direction to the Discussion (section 4.3. Clinical Implications and Future Directions).

  1. Currently, Figure 1 is not very informative—only the inflorescence is shown. I suggest replacing or supplementing it with a botanical illustration of the whole plant, including stem, leaves, roots, and seeds/fruits, to provide a more complete visual reference.

Answer: We followed the reviewer's recommendation and replaced the plant photo in Fig. 1 with the representation of the plant species by Jacob Sturm as pointed in the title of the figure.

  1. More detailed data on the native and current distribution of S. montana would be valuable. Please specify whether the species is considered invasive in certain ecosystems.

Answer: We thank the reviewer for this constructive comment. In response, we have added a brief information regarding the native and current distribution of Satureja montana, and that it is not currently considered invasive in natural ecosystems. This addition can be found in lines 77-79.

  1. The terms "Fidelity Level (FL)" and "Ethnobotanicity Index (EI)" are used only once, and abbreviations are not reused throughout the text. Please choose to either (a) consistently use the full terms or (b) define and use the abbreviations uniformly. Ex.: the same applies to "SM" (Satureja montana), which is inconsistently abbreviated.

Answer: We have taken into account the reviewer's recommendations and revised the text where necessary. Regarding the abbreviation of Satureja montana, it is used along with the full name of the plant, as well as the English name, in order to avoid repetitions given the frequent mention in the text.

  1. All Latin names of species should be written in italic. However, taxonomic families (e.g., Lamiaceae) should remain in regular font.

Answer: We thank the reviewer for this note, the omission has been corrected.

  1. Please add a new column to Table 1 listing the references that reported the presence of the respective compounds.
  2. Consider adding bibliographic references in the same row with the described biological activities for improved traceability.

Answer to 11 & 12: We thank the reviewer for these helpful suggestions aimed at improving clarity and traceability. In response, we have updated Table 1 by adding a new column listing the relevant bibliographic references that confirm the presence of each compound in Satureja montana extracts. Additionally, we have revised Table 2 to include specific references alongside the described biological activities.

  1. Tables 2, 3, and 4 cite sources but use two different formatting styles. I recommend unifying the style of reference citation across all tables.”

Answer: We thank the reviewer for this note, these technical omissions in Tables 2-4 has been corrected.

We express our gratitude to reviewer 3 for the positive assessment of our article, along with appreciation for the time and insightful suggestions that will aid in enhancing the quality of our review.
